# Synthesis, In Vitro Biological Evaluation of Antiproliferative and Neuroprotective Effects and In Silico Studies of Novel 16*E*-Arylidene-5α,6α-epoxyepiandrosterone Derivatives

**DOI:** 10.3390/biomedicines11030812

**Published:** 2023-03-07

**Authors:** Vanessa Brito, Mariana Marques, Marta Esteves, Catarina Serra-Almeida, Gilberto Alves, Paulo Almeida, Liliana Bernardino, Samuel Silvestre

**Affiliations:** 1CICS-UBI—Health Sciences Research Centre, University of Beira Interior, Av. Infante D. Henrique, 6200-506 Covilhã, Portugal; 2Centre for Neurosciences and Cell Biology (CNC), University of Coimbra, 3004-517 Coimbra, Portugal

**Keywords:** 16*E*-arylidene-5α,6α-epoxyepiandrosterone derivatives, epoxidation, aldol condensation, antiproliferative activity, apoptosis, molecular docking

## Abstract

Steroids constitute an important class of pharmacologically active molecules, playing key roles in human physiology. Within this group, 16*E*-arylideneandrostane derivatives have been reported as potent anti-cancer agents for the treatment of leukemia, breast and prostate cancers, and brain tumors. Additionally, 5α,6α-epoxycholesterol is an oxysterol with several biological activities, including regulation of cell proliferation and cholesterol homeostasis. Interestingly, pregnenolone derivatives combining these two modifications were described as potential neuroprotective agents. In this research, novel 16*E*-arylidene-5α,6α-epoxyepiandrosterone derivatives were synthesized from dehydroepiandrosterone by aldol condensation with different aldehydes followed by a diastereoselective 5α,6α-epoxidation. Their cytotoxicity was evaluated on tumoral and non-tumoral cell lines by the 3-(4,5-dimethylthiazol-2-yl)-2,5-diphenyltetrazolium bromide assay. Furthermore, the assessment of the neuroprotective activity of these derivatives was performed in a dopaminergic neuronal cell line (N27), at basal conditions, and in the presence of the neurotoxin 6-hydroxydopamine (6-OHDA). Interestingly, some of these steroids had selective cytotoxic effects in tumoral cell lines, with an IC_50_ of 3.47 µM for the 2,3-dichlorophenyl derivative in the breast cancer cell line (MCF-7). The effects of this functionalized epoxide on cell proliferation (Ki67 staining), cell necrosis (propidium iodide staining), as well as the analysis of the nuclear area and near neighbor distance in MCF-7 cells, were analyzed. From this set of biological studies, strong evidence of the activation of apoptosis was found. In contrast, no significant neuroprotection against 6-OHDA-induced neurotoxicity was observed for the less cytotoxic steroids in N27 cells. Lastly, molecular docking simulations were achieved to verify the potential affinity of these compounds against important targets of steroidal drugs (androgen receptor, estrogen receptor α, and 5α-reductase type 2, 17α-hydroxylase-17,20-lyase and aromatase enzymes). This in silico study predicted a strong affinity between most novel steroidal derivatives and 5α-reductase and 17α-hydroxylase-17,20-lyase enzymes.

## 1. Introduction

Steroids have been considered an important class of pharmacologically active molecules since they play key roles in human physiological processes and are the most important group of regulatory and signaling molecules [1,2]. In humans, the distinct steroids are biosynthesized from cholesterol via enzyme-mediated reactions. Moreover, steroids are lipophilic and readily enter cells, interacting with nuclear and cytosolic receptors and membrane proteins [3]. These molecules have also been associated with several pathological conditions. Considering some relevant properties as low toxicity, less vulnerability to multidrug resistance, and high bioavailability, steroid-based drugs have been a focus of attention to scientific academia and industry for a very long time [4,5,6,7,8]. Therefore, modified steroids comprise a relevant class of bioactive molecules useful for a wide number of diseases (e.g., brain tumors, breast and prostate cancers, fungal and microbial infections, autoimmune disorders), and several representative examples can be found in the literature [2,9,10,11,12,13].

Within this group, steroidal arylidene derivatives, principally those bearing heteroatoms or halogens on their structures, have been mainly reported as 17β-HSD type 1 inhibitors, as well as anti-inflammatory and antiproliferative agents [14,15,16,17,18,19]. Additionally, 16*E*-arylideneandrostane derivatives have been reported as potent anti-cancer agents (a representative example is shown in Figure 1), being potentially useful in the treatment of leukemia, breast and prostate cancer, and brain tumors [15].

On the other hand, epoxysteroids emerged as a class of steroids with several promising bioactivities. As an illustrative example, 5α,6α-epoxycholesterol (Figure 1) is an oxysterol with several biological activities, including regulation of cell proliferation and cholesterol homeostasis [20]. Furthermore, other epoxysteroids have been described in the literature as potential neuroprotective agents, inhibitors of 5α-reductase (5AR), and antiproliferative agents [21,22,23,24]. In this scope, Chávez-Riveros and coworkers reported the synthesis and identification of pregnenolone derivatives as 5AR inhibitors (5ARIs), including an epoxysteroid, 21(*p*-fluoro)benzoyloxy-5α,6α-epoxy-3β-hydroxypregna-16-en-20-one (Figure 1), which presented an IC_50_ value of 179 nM against 5AR type 2 [22]. Interestingly, 21*E*-arylidene-5α,6α-epoxypregnenolone derivatives were already described as potential neuroprotective agents by Jiang and coworkers (a representative example is presented in Figure 1) [21]. Taking in mind all these studies, the modification of different 16*E*-arylidenedehydroepiandrosterones through the stereoselective Δ^5,6^-epoxidation constitutes a valid approach in the research for novel bioactive compounds. 

Thus, in the present study, novel 16*E*-arylidene-5α,6α-epoxyepiandrosterone derivatives were synthesized and biologically evaluated as potential antiproliferative and neuroprotective agents. A battery of several exploratory in vitro assays was performed, including the 3-(4,5-dimethylthiazol-2-yl)-2,5-diphenyltetrazolium bromide (MTT) assay, the immunocytochemistry assay for Ki67 staining, the fluorescence microscopy analysis of propidium iodide (PI) staining, and nuclei morphological analysis, to verify the effects of these novel steroid derivatives on cell proliferation and survival. Furthermore, the possible interactions with the most important typical steroidal targets in this context were studied through molecular docking simulations.

## 2. Materials and Methods

### 2.1. Chemical Synthesis

#### 2.1.1. General Considerations

Reagents and solvents were purchased from commercial suppliers and purified and/or dried whenever necessary using standard procedures before use. The reactions were performed under magnetic stirring using Heidolph plates. Thin layer chromatography (TLC) analysis was performed using 0.20 mm Al-backed silica-gel plates (Macherey-Nagel 60 F254, Duren, Germany), and after elution, plates were visualized under UV radiation (254 nm) in a CN-15.LC UV chamber. Then, the revelation step using an ethanol/sulfuric acid (95:5) mixture followed by heating at 120 °C was performed. Finally, the steroid derivatives were isolated by simple evaporation of solvents using a rotary vacuum dryer from Büchi (R-215). Melting points were measured by a Büchi Melting Point B-540 (Büchi, Switzerland) and were uncorrected. Attenuated total reflectance IR spectra were collected using a Thermo Scientific Nicolet iS10, smart iTR, equipped with a diamond attenuated total reflectance crystal. For IR data acquisition, each solid sample was placed onto the crystal, and the spectrum was recorded. An air spectrum was used as a reference in absorbance calculations. The sample spectra were collected at room temperature in the 4000–600 cm^−1^ range by averaging 16 scans at a spectral resolution of 2 cm^−1^. ^1^H NMR and ^13^C NMR spectra were recorded using a Bruker Avance 400 MHz spectrometer (^1^H NMR at 400.13 MHz and ^13^C NMR at 100.62 MHz) and were processed with the software TOPSPIN (v. 3.1) (Bruker, Fitchburg, WI). Deuterated chloroform (CDCl_3_) was used as a solvent for all samples. Chemical shifts are reported in parts per million (ppm) relative to TMS or solvent as an internal standard. Coupling constants (*J* values) are reported in hertz (Hz), and splitting multiplicities are described as s = singlet, d = doublet, t = triplet, combinations of above, or m = multiplet. ESI-TOF mass spectrometry was performed by the microanalysis service using a QSTAR XL instrument. 

#### 2.1.2. General Procedure to Prepare 16*E*-Arylidenedehydroepiandrosterone Derivatives (**2**–**14**)

An ethanolic solution of dehydroepiandrosterone (DHEA, **1**) (43.2 mg, 1.5 mmol) and aldehyde (1.8 mmol) was added to an aqueous solution of potassium hydroxide (800 μL, 50% *m*/*m*), and the reaction was stirred for 4–24 h at room temperature. The reaction mixture was worked up first by adding cold water to induce precipitation and then filtering and washing the collected solid to obtain the 16*E*-arylideneepiandrosterone derivatives **2**–**14**.


*(E)-16-Benzylidene-3-hydroxy-10,13-dimethyl-1,3,4,7,8,9,10,11,12,13,15,16-dodecahydro-2H-cyclopenta[α]phenanthren-17(14H)-one (**2**)*


White powder (89%); m.p. 179–180 °C; ^1^H NMR (CDCl_3_ 400.13 MHz) δ: 7.47 (2H, d, *J* = 7.2 Hz), 7.39–7.27 (4H, m), 5.33 (1H, s), 3.51–3.42 (1H, m), 2.82 (1H, dd, *J* = 15.8, 6.5 Hz), 1.00 (3H, s), and 0.91 (3H, s). 


*(E)-16-(4-Metoxybenzylidene)-3-hydroxy-10,13-dimethyl-1,3,4,7,8,9,10,11,12,13,15,16-dodecahydro-2H-cyclopenta[α*
*]phenanthren-17(14H)-one (**3**)*


White powder (91%); m.p. 207–208 °C; ^1^H NMR (CDCl_3_ 400.13 MHz) δ: 7.48 (2H, d, *J* = 8.8 Hz), 7.38 (1H, s), 6.92 (2H, d, *J =* 8.9 Hz), 5.38 (1H, s); 3.82 (3H, s), 3.56–3.46 (1H, m), 2.84 (1H, dd, *J* = 15.8, 6.5 Hz), 1.06 (3H, s), and 0.95 (3H, s). 


*(E)-16-(4-Nitrobenzylidene)-3-hydroxy-10,13-dimethyl-1,3,4,7,8,9,10,11,12,13,15,16-dodecahydro-2H-cyclopenta[α]phenanthren-17(14H)-one (**4**)*


White powder (93%); m.p. 250–251 °C; ^1^H NMR (CDCl_3_ 400.13 MHz) δ: 8.24 (2H, d, *J* = 8.6 Hz), 7.64 (2H, d, *J =* 8.5), 7.43 (1H, s), 5.38 (1H, s); 3.57–3.47 (1H, m), 2.86 (1H, dd, *J* = 16.1, 6.5 Hz), 1.06 (3H, s), and 0.98 (3H, s). 


*(E)-16-(Thiophen-2-ylmethylene)-3-hydroxy-10,13-dimethyl-1,3,4,7,8,9,10,11,12,13,15,16-dodecahydro-2H-cyclopenta[α]phenanthren-17(14H)-one (**5**)*


White powder (91%); m.p. 216–218 °C; ^1^H NMR (CDCl_3_ 400.13 MHz) δ: 7.60 (1H, s), 7.49 (1H, d, *J* = 5.1 Hz), 7.31 (1H, d, *J =* 3.5), 7.12–7.08 (1H, m), 5.39 (1H, s); 3.57–3.47 (1H, m), 2.86 (1H, dd, *J* = 16.1, 6.6 Hz), 1.05 (3H, s), and 0.93 (3H, s). 


*(E)-16-(2,3-Dichlorobenzylidene)-3-hydroxy-10,13-dimethyl-1,3,4,7,8,9,10,11,12,13,15,16-dodecahydro-2H-cyclopenta[α]phenanthren-17(14H)-one (**6**)*


White powder (95%); m.p. 137–138 °C; ^1^H NMR (CDCl_3_ 400.13 MHz) δ: 7.75 (1H, s), 7.50–7.40 (2H, m), 7.27 (1H, d, *J =* 8.8 Hz), 5.39 (1H, s); 3.60–3.50 (1H, m), 2.71 (1H, dd, *J* = 15.9, 6.3 Hz), 1.08 (3H, s), and 1.01 (3H, s). 


*(E)-16-(Furan-2-ylmethylene)-3-hydroxy-10,13-dimethyl-1,3,4,7,8,9,10,11,12,13,15,16-dodecahydro-2H-cyclopenta[α]phenanthren-17(14H)-one (**7**)*


White powder (93%); m.p. 184–185 °C; ^1^H NMR (CDCl_3_ 400.13 MHz) δ: 7.54 (1H, s), 7.19–7.16 (1H, m), 6.62 (1H, d, *J =* 3.4 Hz), 6.50–6.47 (1H, m); 5.39 (1H, s), 3.57–3.46 (1H, m), 3.00 (1H, dd, *J* = 16.2, 6.7 Hz), 1.04 (3H, s), and 0.92 (3H, s). 


*(E)-16-[(5-Methylfuran-2-yl)methylene]-3-hydroxy-10,13-dimethyl-1,3,4,7,8,9,10,11,12,13,15,16-dodecahydro-2H-cyclopenta[α]phenanthren-17(14H)-one (**8**)*


Orange powder (88%); m.p. 139–140 °C; ^1^H NMR (CDCl_3_ 400.13 MHz) δ: 7.13 (1H, s), 6.54 (1H, d, *J =* 3.1 Hz), 6.09 (1H, s), 5.39 (1H, s), 3.58–3.46 (1H, m), 2.94 (1H, dd, *J* = 16.3, 6.4 Hz), 1.04 (3H, s), and 0.91 (3H, s).


*(E)-16-(2,4-Dichlorobenzylidene)-3-hydroxy-10,13-dimethyl-1,3,4,7,8,9,10,11,12,13,15,16-dodecahydro-2H-cyclopenta[α]phenanthren-17(14H)-one (**9**)*


Yellow powder (93%); m.p. 192–199 °C; ^1^H NMR (CDCl_3_ 400.13 MHz) δ: 7.71 (1H, s), 7.49 (2H, d, *J =* 8.5 Hz), 7.30 (1H, d, *J* = 8.3 Hz), 5.39 (1H, s), 3.60–3.50 (1H, m), 2.72 (1H, dd, *J* = 15.9, 6.2 Hz), 1.09 (3H, s), and 1.01 (3H, s).


*(E)-16-(4-Methylbenzylidene)-3-hydroxy-10,13-dimethyl-1,3,4,7,8,9,10,11,12,13,15,16-dodecahydro-2H-cyclopenta[α]phenanthren-17(14H)-one (**10**)*


Yellow powder (71%); m.p. 229–231 °C; ^1^H NMR (CDCl_3_ 400.13 MHz) δ: 7.45–7.39 (3H, m), 7.20 (2H, d, *J =* 7.9 Hz), 5.38 (1H, s), 3.57–3.47 (1H, m), 2.85 (1H, dd, *J* = 15.7, 6.7 Hz), 2.36 (3H, s), 1.05 (3H, s), and 0.96 (3H, s).


*€-16-(2,3-Difluorobenzylidene)-3-hydroxy-10,13-dimethyl-1,3,4,7,8,9,10,11,12,13,15,16-dodecahydro-2H-cyclopenta[α*
*]phenanthren-17(14H)-one (**11**)*


Yellow powder (81%); m.p. 140–141 °C; ^1^H NMR (CDCl_3_ 400.13 MHz) δ: 7.56 (1H, s), 7.26 (1H, d, *J =* 7.3 Hz), 7.19–7.06 (2H, m), 5.35 (1H, s), 3.55–3.45 (1H, m), 2.74 (1H, dd, *J* = 16.2, 6.6 Hz), 1.04 (3H, s), and 0.96 (3H, s).


*(E)-16-(4-Bromobenzylidene)-3-hydroxy-10,13-dimethyl-1,3,4,7,8,9,10,11,12,13,15,16-dodecahydro-2H-cyclopenta[α]phenanthren-17(14H)-one (**12**)*


Yellow powder (89%); m.p. 219–220 °C; ^1^H NMR (CDCl_3_ 400.13 MHz) δ: 7.52 (2H, d, *J =* 8.1 Hz), 7.39–7.32 (3H, m), 5.37 (1H, s), 3.57–3.46 (1H, m), 2.81 (1H, dd, *J* = 15.9, 6.4 Hz), 1.06 (3H, s), and 0.95 (3H, s).


*(E)-16-[(5-Chlorothiophen-2-yl)methylene]-3-hydroxy-10,13-dimethyl-1,3,4,7,8,9,10,11,12,13,15,16-dodecahydro-2H-cyclopenta[α]phenanthren-17(14H)-one (**13**)*


Beige powder (94%); m.p. 216–217 °C; ^1^H NMR (CDCl_3_ 400.13 MHz) δ: 7.44 (1H, s), 7.08 (1H, d, *J =* 3.5 Hz), 6.92 (1H, d, *J =* 3.5 Hz), 5.39 (1H, s), 3.60–3.44 (1H, m), 2.76 (1H, dd, *J* = 16.1, 6.4 Hz), 1.05 (3H, s), and 0.92 (3H, s).


*(E)-16-[(5-Chlorofuran-2-yl)methylene]-3-hydroxy-10,13-dimethyl-1,3,4,7,8,9,10,11,12,13,15,16-dodecahydro-2H-cyclopenta[α*
*]phenanthren-17(14H)-one (**14**)*


Orange powder (90%); m.p. 160–161 °C; ^1^H NMR (CDCl_3_ 400.13 MHz) δ: 7.11 (1H, s), 6.64 (1H, d, *J =* 3.4 Hz), 6.31 (1H, d, *J =* 3.4 Hz), 5.43 (1H, s), 3.62–3.52 (1H, m), 2.98 (1H, dd, *J* = 16.8, 6.4 Hz), 1.09 (3H, s), and 0.96 (3H, s).

#### 2.1.3. General Procedure to Prepare 16*E*-Arylidene-5α,6α-epoxyepiandrosterone Derivatives (**15**–**27**)

The substrates **2**–**14** (0.5 mmol) were dissolved in 4.5 mL of CH_2_Cl_2_. To this solution, magnesium monoperoxyphthalate (MMPP; 13.6 mg; 0.4 mmol) and 250 μL of distillated water were added, and the reactional mixture was stirred at room temperature for 24 h. After the reaction is complete, the mixture is filtered, and the organic solvent is removed under reduced pressure. The white solid was dissolved in diethyl ether, and it was washed with an aqueous solution of Na_2_SO_3_ (10%), a saturated NaHCO_3_ solution, and brine. The organic portion was dried with Na_2_SO_4_, filtered, and dried under reduced pressure. The resultant crude was recrystallized from methanol to afford steroids **15**–**27**.


*(E)-16-Benzylidene-3-hydroxy-5α,6α-epoxy-10,13-dimethyl-1,3,4,7,8,9,10,11,12,13,15,16-dodecahydro-2H-cyclopenta[α]phenanthrene-17(14H)-one (**15**)*


White powder (78%); m.p. 207–208 °C; m.p. literature [207–209 °C]; IR (cm^−1^): 3190, 2943, 2859, 1725, 1628. ^1^H NMR (CDCl_3_ 400.13 MHz) δ: 7.51 (2H, d, *J* = 8.1 Hz), 7.42–7.32 (4H, m), 3.94–3.84 (1H, m), 2.95 (1H, d, *J* = 4.4 Hz), 2.85 (1H, dd, *J* = 16.1, 6.4 Hz), 1.11 (3H, s), and 0.90 (3H, s). ^13^C NMR (CDCl_3_ 101.62 MHz) δ: 209.20, 135.58, 135.28, 130.33, 129.30, 128.71, 68.57, 65.87, 65.69, 58.70, 49.95, 47.41, 42.84, 39.76, 35.14, 32.37, 31.20, 31.07, 29.24, 27.90, 20.06, 15.99, 15.29, 14.28 [25].


*(E)-16-(4-Metoxybenzylidene)-3-hydroxy-5α*
*,6α*
*-epoxy-10,13-dimethyl-1,3,4,7,8,9,10,11,12,13,15,16-dodecahydro-2H-cyclopenta[α*
*]phenanthren-17(14H)-one (**16**)*


White powder (33%); m.p. 248–252 °C; IR (cm^−1^): 3517, 2935, 2859, 1725, 1600. ^1^H NMR (CDCl_3_ 400.13 MHz) δ: 7.47 (2H, d, *J* = 8.7 Hz), 7.36 (1H, s), 6.92 (2H, d, *J* = 8.8 Hz), 3.95–3.84 (1H, m), 3.82 (3H, s), 2.94 (1H, dd, *J* = 4.4 Hz), 2.82 (1H, dd, *J* = 15.6, 6.4 Hz), 1.10 (3H, s), and 0.88 (3H, s). ^13^C NMR (CDCl_3_ 101.62 MHz) δ: 209.37, 160.53, 133.20, 133.08, 132.11, 128.27, 114.24, 68.60, 65.71, 58.73, 55.37, 50.04, 47.30, 42.87, 39.79, 35.15, 32.38, 31.22, 31.08, 29.23, 29.08, 27.92, 20.07, 15.99, 14.34. HRMS (ESI-TOF) *m*/*z*: [M+H]^+^ Calcd for C_27_H_34_O_4_ 422.2457; Found 422.2575.


*(E)-16-(4-Nitrobenzylidene)-3-hydroxy-5α*
*,6α*
*-epoxy-10,13-dimethyl-1,3,4,7,8,9,10,11,12,13,15,16-dodecahydro-2H-cyclopenta[α*
*]phenanthren-17(14H)-one (**17**)*


White powder (92%); m.p. 257–258 °C; IR (cm^−1^): 3333, 2937, 2859, 2234, 1717, 1513. ^1^H NMR (CDCl_3_ 400.13 MHz) δ: 8.24 (2H, d, *J* = 8.8 Hz), 7.63 (2H, d, *J* = 8.8 Hz), 7.42 (1H, s), 3.94–3.84 (1H, m), 2.95 (1H, d, *J* = 4.3 Hz), 2.83 (1H, dd, *J* = 16.5, 6.4 Hz), 1.10 (3H, s), and 0.91 (3H, s). ^13^C NMR (CDCl_3_ 101.62 MHz) δ: 208.28, 147.55, 141.91, 139.57, 130.66, 130.38, 123.92, 68.53, 65.66, 58.57, 49.70, 47.56, 42.82, 39.73, 35.12, 32.36, 31.15, 31.06, 29.26, 29.18, 27.88, 20.03, 15.99, 14.21. HRMS (ESI-TOF) *m*/*z*: [M+H]^+^ Calcd for C_26_H_31_NO_5_ 437.2202; found 437.2250.


*(E)-16-(Thiophen-2-ylmethylene)-3-hydroxy-5α*
*,6α*
*-epoxy-10,13-dimethyl-1,3,4,7,8,9,10,11,12,13,15,16-dodecahydro-2H-cyclopenta[α*
*]phenanthren-17(14H)-one (**18**)*


White powder (64%); m.p. 191–192 °C; IR (cm^−1^): 3169, 2942, 2869, 1707, 1670. ^1^H NMR (CDCl_3_ 400.13 MHz) δ: 7.58 (1H, s), 7.49 (1H, d, *J* = 5.0 Hz), 7.31 (1H, d, *J* = 3.4 Hz), 7.13–7.08 (1H, m), 3.95–3.85 (1H, m), 2.96 (1H, d, *J* = 4.4 Hz), 2.85 (1H, dd, *J* = 15.7, 6.6 Hz), 1.10 (3H, s), and 0.86 (3H, s). ^13^C NMR (CDCl_3_ 101.62 MHz) δ: 207.92, 138.86, 132.27, 131.41, 128.68, 126.90, 124.92, 70.58, 67.56, 64.66, 57.69, 48.69, 46.78, 41.84, 38.73, 34.09, 31.38, 30.14, 28.17, 26.86, 19.04, 18.45, 14.98, 13.41. HRMS (ESI-TOF) *m*/*z*: [M+H]^+^ Calcd for C_24_H_30_O_3_S 398.1916; found 398.1985.


*(E)-16-(2,3-Dichlorobenzylidene)-3-hydroxy-5α*
*,6α*
*-epoxy-10,13-dimethyl-1,3,4,7,8,9,10,11,12,13,15,16-dodecahydro-2H-cyclopenta[α*
*]phenanthren-17(14H)-one (**19**)*


White powder (80%); m.p. 135–136 °C; IR (cm^−1^): 3398, 2943, 2856, 2247, 1715, 1626. ^1^H NMR (CDCl_3_ 400.13 MHz) δ: 7.67 (1H, s), 7.41 (1H, d, *J* = 7.8 Hz), 7.36 (1H, d, *J* = 7.8 Hz), 7.21 (1H, t, *J* = 7.9), 3.90–3.80 (1H, m), 2.89 (1H, d, *J* = 15.9, 4.5 Hz), 2.64 (1H, dd, *J* = 6.3 Hz), 1.07 (3H, s), and 0.89 (3H, s). ^13^C NMR (CDCl_3_ 101.62 MHz) δ: 208.09, 138.80, 136.06, 133.92, 133.67, 130.60, 129.41, 127.87, 126.91, 68.53, 65.66, 58.61, 49.74, 47.71, 42.78, 39.74, 35.12, 35.36, 32.36, 31.17, 31.06, 29.22, 28.81, 27.86, 20.02, 15.98, 14.19. HRMS (ESI-TOF) *m*/*z*: [M+H]^+^ Calcd for C_26_H_30_Cl_2_O_3_S 460.1772; found 460.1716.


*(E)-16-(Furan-2-ylmethylene)-3-hydroxy-5α*
*,6α*
*-epoxy-10,13-dimethyl-1,3,4,7,8,9,10,11,12,13,15,16-dodecahydro-2H-cyclopenta[α*
*]phenanthren-17(14H)-one (**20**)*


Yellow powder (92%); m.p. 175–177 °C; IR (cm^−1^): 3162, 2944, 2870, 1708, 1692. ^1^H NMR (CDCl_3_ 400.13 MHz) δ: 7.54 (1H, s), 7.16 (1H, t, *J* = 2.2 Hz), 6.66 (1H, d, *J* = 3.4 Hz), 6.50–6.45 (1H, m), 3.94–3.84 (1H, m), 2.99 (1H, dd, *J* = 15.8, 6.5 Hz), 2.95 (1H, d, *J* = 4.5 Hz), 1.10 (3H, s), and 0.85 (3H, s). ^13^C NMR (CDCl_3_ 101.62 MHz) δ: 209.24, 152.20, 144.80, 133.08, 119.73, 115.81, 112.37, 68.58, 65.75, 58.79, 49.52, 47.53, 42.88, 39.78, 35.14, 32.39, 31.17, 31.07, 29.20, 28.71, 27.91, 20.08, 15.98, 14.38. HRMS (ESI-TOF) *m*/*z*: [M+H]^+^ Calcd for C_24_H_30_O_4_ 382.2244; found 382.2260.


*(E)-16-[(5-Methylfuran-2-yl)methylene]-3-hydroxy-5α*
*,6α*
*-epoxy-10,13-dimethyl-1,3,4,7,8,9,10,11,12,13,15,16-dodecahydro-2H-cyclopenta[α*
*]phenanthren-17(14H)-one (**21**)*


Orange powder (90%); m.p. 153–154 °C; IR (cm^−1^): 2942, 2869, 1708, 1624, 1578. ^1^H NMR (CDCl_3_ 400.13 MHz) δ: 7.06 (1H, s), 6.49 (1H, d, *J* = 3.3 Hz), 6.05 (1H, d, *J* = 3.3 Hz), 3.90–3.80 (1H, m), 2.91 (1H, d, *J* = 4.4 Hz), 2.89 (1H, dd, *J* = 15.1, 6.4 Hz), 2.30 (3H, s), 1.05 (3H, s), and 0.80 (3H, s). ^13^C NMR (CDCl_3_ 101.62 MHz) δ: 209.40, 155.53, 150.74, 131.35, 119.99, 117.51, 109.01, 68.59, 65.84, 60.43, 58.84, 49.65, 47.51, 42.88, 39.75, 35.14, 32.38, 31.18, 31.04, 29.17, 28.67, 27.91, 20.08, 15.99, 14.42, 14.10. HRMS (ESI-TOF) *m*/*z*: [M+H]^+^ Calcd for C_25_H_32_O_4_ 396.2300; found 396.2297.


*(E)-16-(2,4-Dichlorobenzylidene)-3-hydroxy-5α*
*,6α*
*-epoxy-10,13-dimethyl-1,3,4,7,8,9,10,11,12,13,15,16-dodecahydro-2H-cyclopenta[α*
*]phenanthren-17(14H)-one (**22**)*


Yellow powder (91%); m.p. 140–141 °C; IR (cm^−1^): 3418, 2937, 2860, 2245, 1720, 1628, 1582. ^1^H NMR (CDCl_3_ 400.13 MHz) δ: 7.66 (1H, s), 7.46–7.39 (2H, m), 7.29–7.25 (1H, m), 3.94–3.84 (1H, m), 2.92 (1H, d, *J* = 4.4 Hz), 2.66 (1H, dd, *J* = 16.0, 6.3 Hz), 1.09 (3H, s), and 0.90 (3H, s). ^13^C NMR (CDCl_3_ 101.62 MHz) δ: 208.16, 138.25, 136.46, 135.32, 130.47, 129.97, 127.01, 68.55, 65.70, 60.43, 58.62, 49.80, 47.63, 42.77, 39.71, 35.13, 32.35, 31.16, 31.03, 29.20, 27.84, 20.02, 15.98, 14.19. HRMS (ESI-TOF) *m*/*z*: [M+H]^+^ Calcd for C_26_H_30_Cl_2_O_3_S 460.1572; found 460.1605.


*(E)-16-(4-Methylbenzylidene)-3-hydroxy-5α*
*,6α*
*-epoxy-10,13-dimethyl-1,3,4,7,8,9,10,11,12,13,15,16-dodecahydro-2H-cyclopenta[α*
*]phenanthren-17(14H)-one (**23**)*


Pallid yellow powder (92%); m.p. 228–230 °C; IR (cm^−1^): 3227, 2943, 2861, 1714, 1631, 1608. ^1^H NMR (CDCl_3_ 400.13 MHz) δ: 7.39–7.32 (3H, m), 7.15 (12H, d, *J* = 7.7 Hz), 3.90–3.80 (1H, m), 2.89 (1H, d, *J* = 4.3 Hz), 2.79 (1H, dd, *J* = 15.7, 6.3 Hz), 2.32 (3H, s), 1.06 (3H, s), and 0.84 (3H, s). ^13^C NMR (CDCl_3_ 101.62 MHz) δ: 209.45, 139.71, 134.60, 133.38, 132.72, 130.37, 129.47, 68.54, 65.79, 58.76, 49.99, 42.40, 42.86, 40.88, 39.75, 35.13, 32.37, 31.19, 31.03, 29.22, 27.87, 21.49, 20.06, 15.98, 14.30. HRMS (ESI-TOF) *m*/*z*: [M+H]^+^ Calcd for C_27_H_34_O_3_ 406.2507; found 406.2557.


*(E)-16-(2,3-Difluorobenzylidene)-3-hydroxy-5α*
*,6α*
*-epoxy-10,13-dimethyl-1,3,4,7,8,9,10,11,12,13,15,16-dodecahydro-2H-cyclopenta[α*
*]phenanthren-17(14H)-one (**24**)*


Yellow powder (71%); m.p. 140–141 °C; IR (cm^−1^): 3549, 3370, 3225, 2933, 1720, 16301. ^1^H NMR (CDCl_3_ 400 MHz) δ: 7.53 (1H, s), 7.22 (1H, d, *J* = 6.1 Hz), 7.18–7.05 (1H, m), 3.92–3.82 (1H, m), 2.92 (1H, d, *J* = 4.4 Hz), 2.70 (1H, dd, *J* = 16.0, 5.9 Hz), 1.09 (3H, s), and 0.89 (3H, s). ^13^C NMR (CDCl_3_ 101.62 MHz) δ: 208.33, 138.79, 125.80, 124.75, 124.18, 123.85, 120.73, 117.99, 117.79, 68.50, 65.84, 58.70, 49.56, 47.60, 42.80, 39.68, 35.13, 32.33, 31.15, 30.99, 29.20, 29.09, 27.85, 20.03, 15.95, 14.19. HRMS (ESI-TOF) *m*/*z*: [M+H]^+^ Calcd for C_26_H_30_F_2_O_3_ 428.2163; found 428.2239.


*(E)-16-(4-Bromobenzylidene)-3-hydroxy-5α*
*,6α*
*-epoxy-10,13-dimethyl-1,3,4,7,8,9,10,11,12,13,15,16-dodecahydro-2H-cyclopenta[α*
*]phenanthren-17(14H)-one (**25**)*


Yellow powder (77%); m.p. 192–193 °C; IR (cm^−1^): 3226, 2942, 2861, 1715, 1637, 1586. ^1^H NMR (CDCl_3_ 400.13 MHz) δ: 7.50 (2H, d, *J* = 8.5 Hz), 7.33 (1H, d, *J* = 8.5 Hz), 7.30 (1H, s), 3.90–3.80 (1H, m), 2.92 (1H, d, *J* = 4.4 Hz), 2.76 (1H, dd, *J* = 16.0, 6.5 Hz), 1.08 (3H, s), and 0.86 (3H, s). ^13^C NMR (CDCl_3_ 101.62 MHz) δ: 209.04, 136.20, 134.42, 132.02, 131.91, 131.62, 123.54, 69.40, 65.86, 58.74, 49.82, 47.49, 42.80, 39.68, 35.17, 32.40, 31.17, 30.95, 29.21, 27.86, 20.06, 15.97, 14.23. HRMS (ESI-TOF) *m*/*z*: [M+H]^+^ Calcd for C_26_H_31_BrO_3_ 470.1456; found 470.1471.


*(E)-16-[(5-Chlorothiophen-2-yl)methylene]-3-hydroxy-5α*
*,6α*
*-epoxy-10,13-dimethyl-1,3,4,7,8,9,10,11,12,13,15,16-dodecahydro-2H-cyclopenta[α*
*]phenanthren-17(14H)-one (**26**)*


Beige powder (91%); m.p. 119–120 °C; IR (cm^−1^): 3288, 2933, 2859, 2245, 1709, 1680. ^1^H NMR (CDCl_3_ 400.13 MHz) δ: 7.40 (1H, s), 7.05 (1H, d, *J* = 3.9 Hz), 6.90 (1H, d, *J* = 3.9 Hz), 3.90–3.80 (1H, m), 2.94 (1H, d, *J* = 4.4 Hz), 2.71 (1H, dd, *J* = 16.0, 6.5 Hz), 1.07 (3H, s), and 0.82 (3H, s). ^13^C NMR (CDCl_3_ 101.62 MHz) δ: 208.75, 138.65, 134.56, 133.40, 131.82, 127.16, 125.58, 68.43, 65.84, 58.73, 49.68, 47.81, 42.85, 39.69, 35.15, 32.35, 31.14, 30.95, 29.17, 28.66, 27.85, 19.99, 15.95, 14.39. HRMS (ESI-TOF) *m*/*z*: [M+H]^+^ Calcd for C_24_H_39_ClO_3_S 432.1525; found 432.153.


*(E)-16-[(5-Chlorofuran-2-yl)methylene]-3-hydroxy-5α*
*,6α*
*-epoxy-10,13-dimethyl-1,3,4,7,8,9,10,11,12,13,15,16-dodecahydro-2H-cyclopenta[α*
*]phenanthren-17(14H)-one (**27**)*


Pallid powder (72%); m.p. 185–187 °C; IR (cm^−1^): 3400, 2940, 2850, 2246, 1710, 1624. ^1^H NMR (CDCl_3_ 400.13 MHz) δ: 6.70 (1H, s), 6.54 (1H, d, *J* = 3.4 Hz), 6.23 (1H, d, *J* = 3.4 Hz), 3.90–3.80 (1H, m), 2.92 (1H, d, *J* = 4.2 Hz), 1.05 (3H, s), and 0.80 (3H, s). ^13^C NMR (CDCl_3_ 101.62 MHz) δ: 208.98, 151.73, 139.19, 133.48, 118.67, 117.55, 109.23, 68.58, 65.74, 58.79, 58.46, 49.49, 47.53, 42.87, 39.77, 35.13, 32.40, 31.05, 29.16, 28.60, 27.88, 20.05, 18.44, 15.94, 14.31. HRMS (ESI-TOF) *m*/*z*: [M+H]^+^ Calcd for C_24_H_29_ClO_4_ 416.1754; found 416.1785.

### 2.2. Biological Evaluation

#### 2.2.1. Cell Culture

LNCaP, PC-3, MCF-7, PNT1A, N27, and NHDF cells were obtained from the American Type Culture Collection (ATCC; Manassas, VA, USA) and were cultured in 75 cm^2^ culture flasks at 37 °C in a humidified air incubator with 5% CO_2_. LNCaP, PC-3, PNT1A, and N27 cells, respectively, were cultured in RPMI 1640 medium (Sigma-Aldrich, Inc. St. Louis, MO, USA) with 10% fetal bovine serum (FBS; Sigma-Aldrich, Inc. St. Louis, MO, USA) and 1% of the antibiotic mixture of 10,000 U/mL penicillin G and 100 mg/mL streptomycin (Sp, Sigma-Aldrich, Inc. St. Louis, MO, USA). MCF-7 cells were cultured in DMEM medium (Sigma-Aldrich, Inc. St. Louis, MO, USA) supplemented with 10% fetal bovine serum (FBS; Sigma-Aldrich, Inc. St. Louis, MO, USA) and 1% antibiotic/antimycotic (10,000 U/mL penicillin G, 100 mg/mL streptomycin, and 25 µg/mL amphotericin B: Ab, Sigma-Aldrich, Inc. St. Louis, MO, USA). Finally, NHDF cells were cultured in RPMI 1640 medium supplemented with 10% FBS, 2 mM *L*-glutamine, 10 mM HEPES, 1 mM sodium pyruvate, and 1% of the antibiotic/antimycotic Ab, and these cells were used in passages 10th to 14th. For all cell types, the medium was renewed every 2 days until cells nearly reached the confluence state. When cells reach approximately 90–95% confluence, they are gently detached by trypsinization (trypsin-EDTA solution: 0.125 g/L of trypsin and 0.02 g/L of EDTA). Before each experiment, viable cells were counted in a Neubauer chamber by a trypan-blue exclusion assay and adequately diluted in the appropriate complete cell culture medium.

#### 2.2.2. Preparation of Compounds Solutions

All tested compounds were dissolved in dimethyl sulfoxide (DMSO; Sigma-Aldrich, Inc. St. Louis, MO, USA) at a concentration of 10 mM and stored at 4 °C. From the mother-solutions, the diluted solutions of the compounds, in different concentrations, were prepared in the respective complete culture medium before each experiment. The maximum level of DMSO concentration in the studies was 1%, and previous studies demonstrated that this concentration has no relevant effects on cell proliferation (data not shown). 

#### 2.2.3. MTT Cell Proliferation Assay

After reaching a confluence state, cells were trypsinized and counted by the trypan-blue exclusion assay and then seeded with an initial density of 2 × 10^4^ cells/mL in 96-well culture plates (Nunc, Apogent, Denmark) and left to adhere and grow for 48 h. Subsequently, the medium was removed, and the cells were treated with compound solutions (10 and 50 µM for preliminary studies and 0.01, 0.1, 1, 10, 50, and 100 µM for concentration-response studies) in complete culture medium for 72 h. 5-Fluorouracil (5-FU) was used as a positive control, and untreated cells were used as the negative control. Compounds leading to cell proliferation relative to control clearly below 50%, at 10 µM, or clearly below 25%, at 50 µM, in the screening studies were selected to concentration-response studies in tumoral cell lines. Each experiment was performed in quadruplicate and independently at least two times. The in vitro antiproliferative effects were evaluated by the MTT assay (Sigma-Aldrich, Inc. St. Louis, MO, USA). After the incubation period, the medium was removed, and 100 µL of phosphate buffer saline (PBS, NaCl 137 mM; KCl 2.7 mM; Na_2_HPO_4_ 10 mM; and KH_2_PO_4_ 1.8 mM in deionized water with a pH adjusted to 7.4) were used to wash the cells. Then, 100 µL of the MTT solution (5 mg/mL) was prepared in the appropriate serum-free medium and added to each well, followed by incubation for 4 h at 37 °C. Hereafter, the MTT containing medium was removed, and the formazan crystals were dissolved in DMSO. Then, the absorbance was measured at 570 nm using a xMarkTM microplate spectrophotometer from BIO-RAD Laboratories. Cell viability values were expressed as percentages relative to the negative control.

#### 2.2.4. Immunocytochemistry Assay

MCF-7 cells were trypsinized and counted by the trypan-blue exclusion assay, and then cells were seeded with an initial density of 3 × 10^4^ cells/mL in 24-well culture plates (Nunc, Apogent, Denmark) containing circular coverslips of 10 mm diameter. Following seeding (48 h), cells were treated with 1 and 10 µM concentrations of 5-FU, as positive controls, and steroid **19** for 48 h. Untreated cells were used as a negative control. After this period, photograph shots were acquired at an optical microscope coupled with a digital camera (Appendix A). Subsequently, the medium was removed, and cells were fixed with 10% formalin (15 min, room temperature). After fixation, cells were washed three times with PBS, permeabilized, and blocked for non-specific binding sites for 1 h with 0.5% Triton X-100 and 6% BSA at room temperature. Cells were incubated at 4 °C with 30 μL/coverslip of primary antibody, rabbit polyclonal anti-Ki67 (1:50, Abcam Plc.), prepared in 0.3% BSA and 0.1% Triton X-100, overnight. Then, cells were washed three times with PBS and incubated for 1 h, at room temperature, with 30 μL/coverslip of a solution containing the secondary antibody, Alexa Fluor 488 donkey anti-rabbit (1:200, Life Technologies), and the nuclear marker Hoechst-33342 (1:500, Life Technologies) prepared in PBS. Lastly, the coverslips were washed three times with PBS, mounted on a drop of the fluoroshield mounting medium (Abcam, Plc.) on a microscope slide, and left to dry for 24 h. Photomicrographs were taken using an Axio Imager A1 microscope (Carl Zeiss, Göttingen, Germany) with the 40 × objective.

#### 2.2.5. PI Incorporation

After 48 h of being seeded at the same conditions explained in the previous (Section 2.2.4), MCF-7 cells were treated with the steroidal derivatives **19** and 5-FU at 1 and 10 µM and maintained in culture for 48 h. Untreated cells were also used as a negative control. Subsequently, cells were incubated with 20 µL/per well of a propidium iodide solution (PI; 1 mg/mL in PBS; Sigma-Aldrich Co. LLC) for 25 min at 37 °C. Then, the medium was removed, and the cells were fixed with 10% formalin for 15 min at room temperature. Cells were washed three times with PBS and incubated with 30 µL/per coverslip of Hoechst-33342 (1:500) for 10 min. After incubation, cells were washed three times with PBS. Coverslips were mounted on a drop of the fluoroshield mounting medium on a microscope slide, left to dry for 24 h, and then visualized in a Zeiss Axio Imager A1 microscope with the 40× objective.

#### 2.2.6. Analysis of Cell Nuclear Morphology and Distribution with ImageJ

The nuclear measurements were achieved by converting 16-bit photomicrographs of Hoechst 33342-stained nuclei in different studied conditions into 8-bit images. Then, these imagens were autothresholded to binary photos using the default “Make binary” function of ImageJ (v. 1.49) software (NIH Image, Bethesda, MD, USA). Cell nuclei that are touching were separated and fragments were discarded based on area through “Analyze Particle” function. This function also provides several other information, such as nuclear area (NA), circumference, and form factor [26]. Furthermore, the “Nearest Neighbor Distance” (NND) was determined. This function allows measuring the distance between each cell nucleus and the nearest ones, providing information about cell distribution.

#### 2.2.7. Neuroprotective Studies

The immortalized rat mesencephalic dopaminergic cell line (N27 cells; a kind gift from Dr. Yoon-Seong Kim, Burnett School of Biomedical Sciences, University of Central Florida) was grown in Roswell Park Memorial Institute (RPMI) 1640 medium (Sigma-Aldrich) containing 2 g/L sodium bicarbonate, 10% fetal bovine serum (FBS; Millipore), and 1 mL/L of penicillin/streptomycin (GIBCO) in a humidified atmosphere of 5% CO_2_ at 37 °C. Cells were plated at a density of 0.31 × 10^5^ cells per cm^2^ in 96-well culture plates. In the MTT assay, N27 cells were incubated in the presence of several compounds with 50 μM of 6-OHDA. To assess cell viability, MTT reduction was performed as previously described (Section 2.3) with some modifications [27]. Briefly, after 24 h of cell treatments, 0.5 mg/mL of MTT was added to cells for 4 h at 37 °C. Then, the resultant precipitate was dissolved in 10% SDS and quantified at a wavelength of 570 nm.

#### 2.2.8. Statistical Analysis

The MTT results are representative of at least two independent experiments (N ≥ 2), each comprising one sample with four replicates (*n* = 4). The data are expressed as mean ± standard deviation (SD). The analysis of immunocytochemistry and cell nuclear morphology and distribution experiments were performed at the border of the coverslips, where cells formed a pseudo monolayer. The experiments were performed in at least two independent cultures (N ≥ 2, n = 2), and the results are expressed as mean ± standard error of the mean (SEM). The results of the effects on cell survival with 6-OHDA determined by the MTT assay are expressed as mean ± SEM. The data are representative of three independent experiments (N = 3), conducted in duplicate (*n* = 2). Except for the neuroprotection study, all statistical significances were determined with GraphPad Prism 6 software (GraphPad, San Diego, CA, USA) by using ANOVA followed by Dunnett’s or Bonferroni’s tests or an unpaired, two tailed Student *t*-test. For the neuroprotection assay, statistical analysis was determined with GraphPad Prism 9 software by using a one-way ANOVA followed by a Tukey multiple comparison test. Differences between groups were considered statistically significant at a *p*-value lower than 0.05 (*p* < 0.05).

### 2.3. Molecular Docking Simulations

#### 2.3.1. Preparation of Macromolecules and Ligands

The 3D structural coordinates of 5α-reductase type II (5AR PDB code: 7BW1), estrogen receptor-α (ERα PDB code: 1A52), androgen receptor (AR PDB code: 2AMA), 17α-hydroxylase-17,20-lyase (CYP17A1 PDB code: 3RUK), and aromatase (PDB code: 3EQM) were retrieved from the Protein Data Bank site (https://www.rcsb.org; accessed on 30 August 2022). The coordinates of the co-crystallized ligands and water molecules were deleted using the software Chimera (v. 1.10.1), histidine charges were defined to match the physiologic environment, and the final structures were saved in PDB format. Then, non-polar hydrogens were merged in AutoDockTools (v. 1.5.6) from the Scripps Research Institute [28]. Kollman and Gasteiger partial charges were added to the same software. Lastly, the prepared structures were converted from the PDB format to PDBQT. All ligands were drawn using Chem3D (v. 22.0 Free Trial) software (by Cambridge ChemBioOffice 2010). Then the energies were minimized and the geometry was optimized (MMFF94 force field: 500 steps of conjugate gradient energy minimization followed by 500 steps of steepest descent energy minimization with a convergence setting of 10 × 10^−7^) with the software Avogadro (v. 1.0.1), and the final structures were saved as PDB file format. The ligands were completely prepared by choosing torsions, and the structures were converted from PDB format to PDBQT in AutoDockTools.

#### 2.3.2. Grid Map Parameters

The grid parameters were determined using AutoDockTools based on the coordinates of the ligand crystallized from each complex macromolecule-ligand, namely finasteride, estradiol, DHT, abiraterone, and androstenedione, with the respective macromolecule. The grid box was centered on the ligand with the following coordinates: 5AR coordinates were x = −27.450, y = 15.112, z = 31.795; ERα coordinates were x = 90.36, y = 13.681, z = 72.011; AR coordinates were x = 27.603, y= 1.834, z = 4.722; CYP17A1 coordinates were x = 27.256, y = −0.978, z = 33.104 and Aromatase coordinates were x = 86.071, y = 54.241, z = 46.085. The size of the grid box was 20 × 20 × 20 with a spacing of 1.0 Å.

#### 2.3.3. Method Validation and Molecular Docking Simulations

Scoring functions are essential for molecular docking performance. In order to verify those functions, it is necessary to validate the docking performance of AutoDock Vina. This step is required to verify the performance by analyzing the difference between the real and best-scored conformations. For the docking process to be considered successful, the root-mean-square distance (RMSD) value between those two conformations must be less than 2.0 Å. In this case, the validation method was performed by re-docking 5AR with the nicotinamide-adenine-dinucleotide phosphate-dihydrofinasteride (NADP-DHF) adduct, ERα with estradiol, AR with DHT, CYP17A1 with abiraterone, and Aromatase with androstenedione. Low RMSD values were obtained for all cases, which means that the docking process was reliable and validated. After ligands and protein preparation, as well as method validation, molecular docking was performed by the AutoDock Vina executable, which uses an iterated local search global optimizer. The parameter of exhaustiveness of the performed experiments was defined as 15. The results of molecular docking were visualized in the Discovery Studio Visualizer (v. 5.0) program from BIOVIA and in PyMOL (v. 1.1) software.

## 3. Results and Discussion

### 3.1. Chemical Synthesis

The synthesis of novel 16*E*-arylidene-5α,6α-epoxyepiandrosterone derivatives has been carried out as shown in Figure 1. Initially, the preparation of 16*E*-arylidene-epiandrosterone derivatives (**2**–**14**) was accomplished through a crossed-aldol condensation, named the Claisen-Schmidt reaction, by mixing a solution of DHEA (**1**) and different aromatic aldehydes in EtOH with an aqueous solution of KOH (50%). After 2–24 h at room temperature, the intermediate products were obtained in very good to excellent yields (71–95%) [29]. Since this type of intermediate is already described in the literature [15,30,31,32,33,34], a basic structural characterization (^1^H NMR spectrum acquisition and m.p. determination) was performed to prove their obtainment. Then, based on a procedure described by Carvalho et al., several new epoxysteroids (**15**–**27**) were obtained from substrates **2**–**14** by a stereoselective reaction with the peroxyacid MMPP [20]. Overall, epoxidation of steroids with *trans-anti-trans* ring fusions generally leads to the formation of the α-epoxide. This fact can be explained by the preference of the attack by the reagent on the α-side of the steroid scaffold since the β-side is shielded by the two angular methyl groups at C-10 and C-13 [35]. Despite 5α,6α-epoxysteroids have generally been obtained by oxidation with the most common peroxyacid 3-chloroperoxybenzoic acid (*m*-CPBA), MMPP has emerged as an advantageous alternative, considering its higher stability in the solid state and safety in handling. In addition, this peroxyacid can be produced and commercialized at a lower cost, and the respective work-up procedures are relatively easy due to its water solubility [20,36,37,38]. Moreover, MMPP is more selective than *m*-CPBA considering the obtention of α-epoxides, leading to significantly higher values of α/β epoxide ratios [20,39]. To improve this ratio in the final products (**15**–**27**), a recrystallization with methanol was performed, but only very small amounts of the β-epoxide (<2%) could be detected in the ^1^H NMR spectra for these epoxysteroids.

In general, these novel 16*E*-arylidene-5α,6α-epoxyepiandrosterone derivatives (**15**–**27**) were synthesized with reasonable overall yields (30–86%) (Table 1). Interestingly, discrepant yields were achieved for these steroidal derivatives, suggesting different reactivity considering the substituent present at C-16 (the aromatic group). Relative to structural characterization, with the exception of epoxide **15** [25], steroidal epoxides **16**–**27** are new synthetic molecules, and respective high resolution mass spectral acquisition (HRMS) was mandatory, as well as IR and ^13^C NMR spectra (data available on Appendix A). 

### 3.2. Biological Evaluation

#### 3.2.1. Antiproliferative Activity

The effects of epoxysteroids **15**–**27** on the proliferation of non-tumoral cells and tumoral cells were assessed by the MTT proliferation colorimetric assay. PNT1A (normal prostate cells), NHDF (normal human dermal fibroblasts), and N27 (rat dopaminergic neuronal cells) were used as models of non-tumoral cells. As models of androgen-dependent and androgen-independent prostatic cancer, LNCaP and PC-3 cell lines were used, respectively. MCF-7 cells are estrogen-responsive breast cancer cells. In the first assay, cells were exposed to all of these new steroidal derivatives at concentrations of 10 and 50 μM for 72 h. Additionally, 5-FU was also introduced in the assay as a clinically used antitumor positive control. The results of this preliminary study are depicted in Figure 2.

From this preliminary study, compounds **20** and **21** seemed to not display a strong effect on cell proliferation against tumoral cell lines (relative cell proliferation ≥50% at both concentrations), being apparently more cytotoxic against non-tumoral cells, principally with steroid **21**. On the other hand, steroids **15**, **19**, **22**, **23**, **25**, **26**, and **27** showed strong effects on cell proliferation, principally at 50 μM (relative cell proliferation ≤50%), against all cell lines. In addition, at 50 μM, almost all the steroids (**15**, **19**, **20**, **22**, **25**, and **26)** had a strong effect on the proliferation of PC-3 cells, causing a decrease on cell proliferation clearly higher than 75% relative to the control. Similarly, steroidal derivatives **19**, **22**, **23**, **25**, **26**, and **27**, at 50 µM, also produced a decrease in LNCaP cell proliferation greater than 75%. However, at 10 µM, the same steroids in the same cell line seemed to have no relevant effects. Notably, in the MCF-7 cell line, treatment with steroids **19** and **22** at 50 μM reduced cell proliferation to nearly 0% relative to the control. Generally, these epoxides seemed to be the most potent compounds, having significant effects in all cell lines. In summary, after this preliminary screening assay, there is no doubt that steroid **19** produced the most significant effect on cell proliferation, mainly in MCF-7 cells, at the two tested concentrations (10 and 50 µM). In fact, in addition to causing a decrease close to 100% in cell proliferation at 50 µM, it also caused a decrease of around 75% at 10 µM.

After this preliminary evaluation, concentration–response studies were performed for the most antiproliferative compounds (relative cell proliferation <50% at 10 µM or <25% at 50 µM) and 5-FU in tumoral cells. In relation to non-tumoral cells (NHDF, PNT1A, and N27), concentration-response values were determined for all new derivatives and 5-FU to analyze their selectivity (non-tumoral vs. tumoral cells) and to acquire data relevant for the neuroprotective studies (N27 cells). The IC_50_ values were estimated by treating cells with solutions of the tested compounds at concentrations of 0.01, 0.1, 1, 10, 50, and 100 μM for 72 h. As an example, the concentration-response curves for compound **19** in all cell lines are presented in Appendix A. The determined IC_50_ values are shown in Table 2 (the R squared associated with each IC_50_ is presented in Appendix A), and some relevant points can be highlighted. For example, in the androgen-dependent cell line LNCaP, a relative selectivity can be observed for steroid **27**, with an interesting IC_50_ value (15.80 μM), lower than that observed in the other cell lines. However, the lowest values were determined in MCF-7 cells, particularly for epoxide **19,** which is the most potent steroid (IC_50_ = 3.47 μM) identified in the present study. In addition, this value is lower than that determined for 5-FU, the positive control (IC_50_ = 6.30 μM), being the only case concerning tumoral cells. Furthermore, the determined IC_50_ values against non-tumoral cells are higher than the observed values for MCF-7 cells. Consequently, this compound was selected for further studies to better understand the potential mechanisms underlying the cytotoxicity observed.

Generally, comparing the antiproliferative activity of these new steroidal 5α,6α-epoxides with 16*E*-arylidenedehydroepiandrosterone derivatives already described in the literature, it can be observed that epoxysteroids seem to have higher antiproliferative activity against tumoral cell lines [15,30,33,40]. For instance, Vosooghi and coworkers prepared and evaluated against tumoral cell lines by MTT assay the intermediate products herein named as **4** and **9**, and these steroids presented IC_50_ values against T47-D cells (breast cancer) >100 µM. On the contrary, the corresponding epoxysteroids (**17** and **22**, respectively) exhibited lower IC_50_ values against MCF-7 cells (also from breast cancer): IC_50_(**17**) = 20.82 µM and IC_50_(**22**) = 14.46 µM [15]. Despite this, a few 16*E*-arylidenedehydroepiandrosterone derivatives with interesting cytotoxicity can also be highlighted. For example, it was shown that the DHEA derivative bearing a 3-chlorobenzylidene at C16 was the most potent against KB (nasopharyngeal epidermoid carcinoma) and T47D cells (IC_50_ values of 0.6 and 1.7 μM, respectively) [15]. Considering all these results, chemical modifications on these compounds to improve their bioactivity are rationally justified.

#### 3.2.2. Characterization of Steroid 19 Effects on MCF-7 Cells Proliferation 

Considering the MTT proliferation assay results, in which steroid **19** exhibited the most potent effect (IC_50_ = 3.47 μM) in the MCF-7 cell line, a panel of more specific biological experiments with this derivative was accomplished. This set of biological evaluation methodologies comprised an immunocytochemistry assay, PI incorporation, and analysis of cell nuclear morphology and distribution using ImageJ software.

In the present study, the expression of Ki67 was assessed in MCF-7 cells after 48 h of treatment with 5-FU, as a positive control, and steroid **19** at 1 and 10 μM. 5-FU was included in the assay due to its capacity to induce apoptosis. The 5-FU is an analogue of uracil, and consequently, it rapidly enters the cell through the same facilitated transport mechanism. Then, 5-FU is converted intracellularly into active metabolites: fluorodeoxyuridine monophosphate, fluorodeoxyuridine triphosphate, and fluorouridine triphosphate. These active metabolites are able to disrupt RNA synthesis and inhibit the action of thymidylate synthase, leading the cell to apoptosis [41]. The monoclonal antibody Ki67 detects a human nuclear antigen that is present in proliferating cells but absent in quiescent cells [42,43]. In addition, this marker is commonly used to stratify good and poor prognostic categories in invasive breast cancer [44,45]. In a parallel study, MCF-7 cells were stained with PI, and the percentage of PI-positive cells was measured after treatment with 5-FU and **19**. The exclusion of fluorescent dye PI is a widely used assay in tissue culture systems, with PI labeling the nucleus in dying cells with a compromised plasma membrane [46,47]. This method has proved to be particularly useful in studying cell death (necrosis and late apoptosis) in vitro [48,49].

The results of the percentages of Ki67-positive and PI-positive cells (% relative to control) obtained are shown in Figure 3. In addition, representative photomicrographs of MCF-7 cells after Hoechst-33342/Ki67 and Hoechst-33342/PI staining are presented in Figure 4. From these experiments, it was observed that 5-FU and steroid **19**, at 10 μM, induced a statistically significant decrease of Ki67-positive cells relative to control in a very similar manner (*p* < 0.0001). Moreover, at 1 μM, none of the compounds significantly affected the percentage of Ki67-positive cells when compared to the control. Concerning PI staining, again at the lowest concentration, no significant effect was observed on the percentage of PI-positive cells relative to the negative control. On the contrary, at 10 μM, both compounds showed the ability to significantly increase the percentage of PI-positive cells. However, compared with 5-FU (*p* < 0.05), the effect of epoxide **19** is statistically more significant (*p* < 0.0001), increasing about three times the percentage of PI incorporation compared with the positive control.

Additionally, considering cellular morphologic features associated with apoptosis/necrosis, in the early stages of apoptosis, cells preserve organelles and the cell membrane, whereas the nucleus undergoes early degeneration. Usually, cap-shaped chromatin margination is one of the first signs of apoptosis, as are cytosol condensation, pyknosis, and cell membrane blebbing. Then, the nucleus develops several dense and circular micronuclei, which are released into the extracellular space. Lastly, the cells split into numerous apoptotic bodies. In vitro, the late stage of necrosis presented by apoptotic cells is usually characterized by early cell membrane damage. Nevertheless, pyknotic and fragmented cell nuclei are not common in necrosis [50,51,52]. In necrotic cells, the nucleus stays relatively intact while the cell membrane and organelles show early degeneration [50]. Thus, these distinctive modifications in nuclear morphology during apoptosis can be used as markers of this cell death mechanism. In this study, bearing in mind the research reported by Eidet and coworkers, the NA and cell distribution through the NND were analyzed as principal indicators of apoptosis [26]. The results of this study are shown in Figure 5. Interestingly, after exposure to steroid **19** at both tested concentrations, the NA of MCF-7 cells decreased significantly (*p* < 0.0001) relative to the control (Figure 5A). Moreover, this effect is more drastic for epoxide **19** than the observed one for 5-FU, the reference. In relation to NND measurement, a similar effect was verified when cells were treated with 5-FU and the steroid **19** at 1 and 10 μM (Figure 5B). Therefore, the cell distribution analysis indicated a more random distribution with nuclei more spaced after 48 h of incubation with the tested compounds. In this context, Eidet and coworkers demonstrated the relationship between apoptosis activation and more uneven cell spacing [26]. In conclusion, the results obtained from these more specific experiments constitute a strong indication of the possibility of **19** triggering the apoptotic cell death pathway. However, other studies should be performed to confirm this hypothesis.

#### 3.2.3. Study of Neuroprotective Effects on N27 Cells

Jiang and coworkers reported that 21*E*-arylidenepregnenolone derivatives, including 5α,6α-epoxides, are potential neuroprotective agents [21]. Considering their resemblance with the compounds synthesized in the present work, the assessment of the effects of the prepared steroids on the survival of N27 dopaminergic cells exposed to 6-OHDA was determined by the MTT assay. 6-OHDA is a neurotoxin that induces degeneration of catecholaminergic neurons, including dopaminergic neurons, and is frequently used as a toxin-induced model of Parkinson’s disease [53,54]. As steroids **17**, **20**, and **27** revealed lower cytotoxicity against N27 cells (Figure 2 and Table 2), they were tested at 10 and 50 µM. Hence, cells were exposed to 6-OHDA, at 50 µM, together with steroids **17**, **20**, and **27**, for 24 h [55]. In addition, 21*E*-benzylidene-5α,6α-epoxipregnenolone (hereafter denoted as **cD**) was also prepared by us by means of the above described procedures (the spectral data is in agreement with that described in ref. [21]) and used in this study as a representative compound of the 21*E*-arylidenepregnenolone derivatives previously studied in this context [21]. 

The loss of dopaminergic neurons is one of the main hallmarks of Parkinson’s disease, and 6-OHDA can replicate some pathological features of this disease, including dopaminergic neuronal loss [56,57]. N27 cells treated with 50 μM 6-OHDA for 24 h presented an almost 50% reduction in cell viability compared to the control (Figure 6), showing that exposure to this toxin leads to significant dopaminergic cell death. The incubation with 10 µM of **cD** as well as with compounds **17**, **20,** and **27** did not affect cell viability. Nevertheless, N27 cells treated with 50 µM of compounds **17** and **27** presented a fold-increase of 1.53 and 1.25, respectively, in cell viability compared to the control and a fold-increase of 1.59 and 1.41, respectively, compared to 10 µM of the respective compounds, after 24 h exposure (Figure 6). The opposite effect was observed after 72-h of incubation with the same compounds (Figure 2), indicating that compounds **17** and **27** affect N27 cell survival in a time-dependent manner. 

Using different neuronal cell models, Jiang and coworkers showed that a series of 21-arylidenepregnenolone derivatives had neuroprotective effects against amyloid-β25–35-, hydrogen peroxide-, and oxygen-glucose deprivation (OGD)-induced neurotoxicity [21]. Similarly, the neurosteroid pregnenolone was also shown to protect against glutamate and amyloid β-induced toxicity in the HT-22 mouse hippocampal cell line [58]. The protective effect of steroids was further demonstrated in in vivo models. In lipopolysaccharide (LPS)-treated rats and mice, several 16-substituted steroids and 16-arylidene steroidal derivatives improved the locomotion and cognition of the animals and ameliorated LPS-induced neuroinflammation [59,60]. In 1-methyl-4-phenyl-1,2,3,6-tetrahydropyridine (MPTP) rodent models, the treatment with DHEA and 17β-estradiol increased the concentration of catecholamines, including dopamine, protected dopaminergic neurons, and attenuated microglia and astrocytic activation [61,62]. However, testosterone treatment did not protect against MPTP-induced toxicity in the rat striatum [61]. 

In our study, the incubation of N27 cells with 10 or 50 µM **cD** did not protect against 6-OHDA-induced neurotoxicity (Figure 6). However, other derivatives may be tested in future experiments to validate their potential neuroprotective effect against 6-OHDA. In line with this finding, the incubation of N27 cells with compounds **17**, **20**, and **27**, at 10 and 50 µM, did not counteract dopaminergic degeneration induced by 6-OHDA. Moreover, although 50 µM of **27** alone increased the viability of N27 cells, its incubation with 6-OHDA was revealed to be toxic to cells compared to 6-OHDA alone (a 0.9-fold-decrease; Figure 6). This may suggest that their effects are context-dependent.

Altogether, our results do not support a neuroprotective role of 21*E*-arylidenepregnenolone derivatives as well as of these novel epoxysteroids against the toxicity induced by 6-OHDA in N27 dopaminergic cells. 

### 3.3. Molecular Docking Simulations

To evaluate the potential affinity of these novel steroids for several proteins, an in silico study using molecular docking simulations was accomplished. This study aimed to verify the existence of potential interactions between these novel epoxysteroids and proteins that are known to interact with steroids. Thus, these simulations were performed against known targets of steroidal drugs currently used in the treatment of breast cancer, benign prostatic hyperplasia (BPH), and PCa, namely, 5AR, ERα, AR, CYP17A1, and aromatase. The protein targets were selected according to the following criteria: a high-resolution X-ray crystal structure is available in a complex with a steroidal drug or other ligand, and the protein is a target of clinically approved steroid-based anticancer drugs used in the treatment of hormone-dependent breast cancer, or PCa. Moreover, 5AR was included in this assay since the use of 5ARIs, including steroidal drugs (e.g., finasteride), constitutes the main strategy in the treatment and management of BPH. Three-dimensional structural coordinates of protein receptors were obtained from the protein databank (PDB), and molecular docking was performed using the program AutoDockTools. The present study was performed for each new derivative, **15**–**27**, against all mentioned targets. Firstly, the docking method validation was mandatory, and simulations were carried out between crystallized ligands/drugs and the respective proteins. From this validation study, it was observed that all control redocking simulations were able to reproduce the ligand–protein interaction geometries presented in the respective crystal structures with a root-mean-square distance (RMSD) <2.0 Å. On the basis of the control redocking simulations, predicted binding energies are analyzed in comparison with the value observed for the control. 5AR was co-crystallized with the NADP-DHF adduct, and thus the redocking study was performed with 5AR complexed with this adduct. However, an additional simulation using only finasteride and 5AR was achieved to subsequently compare with the novel epoxysteroids. The predicted binding energies of steroids **15**–**27** are shown in Table 3, and some relevant binding energies can be observed. In general, considering the lowest affinity energies, the tested steroids exhibited higher affinity to 5AR and to CYP17A1, and very low affinity to ERα and AR. Regarding the results against 5AR, it is interesting to note that, with the exception of steroid **18**, all epoxysteroids drugs have equal or lower energies than the control, finasteride. These results showed that there is a possibility that these synthesized final products could be potential 5AR inhibitors. Moreover, when the molecular docking was performed with CYP17A1, it was observed that, when compared with abiraterone, the affinity of the tested compounds is generally very similar. These interesting values can be suggestive that these steroidal derivatives can also have relevant interactions with this relevant target, which is important on the androgen signaling pathway in PCa [63,64]. On the other hand, the values obtained against ERα and AR are clearly indicative of the absence of relevant interactions between these targets and the novel epoxysteroid, being energy values that are significantly higher than the control. Given the typical selectivity problem of this type of compound, this result is interesting. Lastly, regarding the results against aromatase, it was observed that all compounds, with the exception of steroid **22**, generated affinity energies higher than the redocking values, suggesting lower affinity than androstenedione (the co-crystallized ligand). Despite epoxide **22** exhibiting slightly higher than androstenedione, probably this difference is not significant to improve the bioactivity. In addition, after the visualization and study of the potential interactions between **22** and aromatase in the Discovery Studio software, it was observed that the most relevant interactions to the enzyme activity are mostly absent (Appendix A). 

## 4. Conclusions

A new series of 16*E*-arylidene-5α,6α-epoxyepiandrosterone derivatives was successfully synthesized with reasonable overall yields. Interestingly, some of these compounds have shown selective cytotoxic effects in tumoral cell lines, with an IC_50_ of 3.47 µM for the 2,3-dichlorophenyl derivative **19** against MCF-7 cells. Furthermore, immunocytochemistry assays after exposure to steroid **19** showed a significant decrease in the cell proliferation marker Ki67 and a significant increase in cell death. Concerning morphologic features, the measurement of the nuclear area revealed a significant decrease after treatment with **19** at both tested concentrations. This effect, triggered by steroid **19**, is more statistically significant than the effect observed for the reference compound 5-FU. In addition, the assessment of cell distribution through the measure of nearest neighbor distance indicated a more aleatory distribution in a similar way after exposure to **19** and 5-FU, the positive control. These results could be strongly indicative of the activation of the apoptotic pathway triggered by epoxide **19**. On the other hand, no significant neuroprotection against 6-OHDA neurotoxicity was observed for the less cytotoxic steroids **17**, **20,** and **27** in N27 cells. Molecular docking simulations suggested, principally, a strong affinity between most of the novel steroidal derivatives and 5AR and a very similar interaction mode in comparison with finasteride. Similarly, these novel steroids seemed to have a strong affinity for CYP17A1. Concluding, several 16*E*-arylidene-5α,6α-epoxyepiandrosterone derivatives with interesting selective antiproliferative effects were successfully prepared, and further studies to assess the potential inhibition of 5AR are in progress.

## Data Availability

The data presented in this study are available in this article and the Appendix A.

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
