# Peer review of "Synthesis, In Vitro Biological Evaluation of Antiproliferative and Neuroprotective Effects and In Silico Studies of Novel 16E-Arylidene-5α,6α-epoxyepiandrosterone Derivatives"

_biomedicines, 2023, doi:10.3390/biomedicines11030812_

Round 1

Reviewer 1 Report

V. Brito et al. describe a series of epoxy-steroids with antiproliferative activities. The authors confirm the antiproliferative effect of the compounds and a possible apoptosis pathway. Then, they show that the compounds do not seem to have a neuroprotective effect, and finally docking studies suggest interaction with 5AR.

The manuscript is well written, and I consider it worth for publication in Biomedicines. However, there are some data that needs to be reviewed for the characterisation of the compounds, and the following aspects could be improved:

-          It would be useful to have a Figure in the introduction showing some of the structures mentioned from previous studies. This would also help assess the difference between the already known compounds and theirs.

-          Many proton and carbon signals are missing in the NMRs. Looking at the NMR spectra included in the supplementary, it looks like the authors have not manually integrated and selected the peaks but rather done it automatically, so the integrations are not covering well the peaks and there are many signals not integrated. This needs to be solved before the manuscript can be published.

-          Why is the MTT protocol explained twice?

-          Figure 1 is a scheme, not a figure.

-          Figure 2 does not indicate what the different colours mean.

-          It would be useful to include some representative dose-response curves in the supplementary information.

-          R2 is not needed in Table 2. It complicates the interpretation.

-          P14, L575: “After this preliminary evaluation, concentration–response studies were performed for the most antiproliferative compounds”. However, Table 2 seems to show concentration-response studies for all compounds.

-          P14, L583: “Steroids 15, 16, and 24 presented some selectivity for PC-3 cells”. However, Table 2 does not show value for 24 in PC-3 cells. Also, for 15 and 16 the differences with the other cell lines are too small to claim selectivity for PC-3 cells.

-          P14, L586 “Relative to steroid 18, the IC50 values are lower in non-tumoral cells [IC50(NHDF)= 29.65 μM; IC50(PNT1A)= 29.56 μM; IC50(N27)= 12.58 μM] than in PC-3 cells (IC50= 37.56 μM).”. These values do not look that different. In general, the authors have to review the discussion on the results as many of these values are claimed different but look quite close.

-          P14, L602. Given the discussion on alkenes vs epoxides, it would be interesting to see the activity of for instance intermediate 6.

-          Figure 4: Caption should indicate what the arrows mean.

-          In references, please review the format of the doi for some references.

Author Response

REVIEWER#1

Brito et al. describe a series of epoxy-steroids with antiproliferative activities. The authors confirm the antiproliferative effect of the compounds and a possible apoptosis pathway. Then, they show that the compounds do not seem to have a neuroprotective effect, and finally docking studies suggest interaction with 5AR.

The manuscript is well written, and I consider it worth for publication in Biomedicines.

Authors’ response: Thank you very much for your positive comments.

However, there are some data that needs to be reviewed for the characterisation of the compounds, and the following aspects could be improved:

-It would be useful to have a Figure in the introduction showing some of the structures mentioned from previous studies. This would also help assess the difference between the already known compounds and theirs.

Authors’ response: Thank you very much for your suggestion. A figure showing some relevant structures from previous studies was incorporated in introduction. As this is a new figure (Fig.1), adjustments in numbering the remaining Figures in the text were performed.

-Many proton and carbon signals are missing in the NMRs. Looking at the NMR spectra included in the supplementary, it looks like the authors have not manually integrated and selected the peaks but rather done it automatically, so the integrations are not covering well the peaks and there are many signals not integrated. This needs to be solved before the manuscript can be published.

Authors’ response: Thank you very much for this pertinent observation. We understand your concern; however the signals of aliphatic protons without integration correspond to the so called “methylenic bloc”, which usually ranges from 0.5 to 2.5 ppm. These signals are originated from aliphatic protons without marked steric and/or electronic influence and thus have high overlapping. Considering this fact, it is only possible to perform the signals attribution to each proton with confidence using advanced NMR techniques and, in some cases, doubts will still persist. Therefore, it is frequent that in 1H-NMR steroids characterization only the signals outside the “methylenic bloc” as well as the signals with higher integration are presented. An example is the following article:

https://www.sciencedirect.com/science/article/pii/S0960076021001904?via%3Dihub#sec0010

Moreover, some authors also present these signals as “overlapping multiplets” integrating diverse numbers of hydrogens. As example, please see:

https://www.sciencedirect.com/science/article/pii/S0960076022000152?via%3Dihub

As we are presenting 1H-NMR spectra of these compounds in “Supplementary Material”, we believe that it is no necessary to include more information concerning this point in the manuscript.

-Why is the MTT protocol explained twice?

Authors’ response: In fact, the Reviewer is correct and we thank you for this observation. Although the MTT assay was carried out in different contexts, the protocol followed was relatively similar. Considering this fact, the text was adapted in section 2.2.7. of the manuscript.

-Figure 1 is a scheme, not a figure.

Authors’ response: Thank you for this correction. It was altered in the manuscript.

-Figure 2 does not indicate what the different colours mean.

Authors’ response: The meaning of the different colors was added to the respective legend.

-It would be useful to include some representative dose-response curves in the supplementary information.

Authors’ response: We appreciated this suggestion and agree with you. Therefore, the concentration-response curves for compound 19 in all cell lines were included in supplementary material.

-R2 is not needed in Table 2. It complicates the interpretation.

Authors’ response: Thank you for this suggestion. Therefore, the R2 value was removed from Table 2 in the manuscript. However a table with this information was included in Supplementary Material (S3).

-P14, L575: “After this preliminary evaluation, concentration–response studies were performed for the most antiproliferative compounds”. However, Table 2 seems to show concentration-response studies for all compounds.

Authors’ response: Considering Table 2, in fact, IC50 values were determined for all compounds in normal cell lines (NHDF, PNT1A and N27) in other to compare their potency against non-tumoral vs tumoral cells (selectivity). To clarify this question, the text was adjusted in the manuscript.

-P14, L583: “Steroids 15, 16, and 24 presented some selectivity for PC-3 cells”. However, Table 2 does not show value for 24 in PC-3 cells. Also, for 15 and 16 the differences with the other cell lines are too small to claim selectivity for PC-3 cells.

Authors’ response: Thank you for these observations. We agree with you and, accordingly, we modified the text in this section to reduce the possibility of results misinterpretations.

-P14, L586 “Relative to steroid 18, the IC50 values are lower in non-tumoral cells [IC50(NHDF)= 29.65 μM; IC50(PNT1A)= 29.56 μM; IC50(N27)= 12.58 μM] than in PC-3 cells (IC50= 37.56 μM).”. These values do not look that different. In general, the authors have to review the discussion on the results as many of these values are claimed different but look quite close.

Authors’ response: Thank you for this observation. In fact, we agree with the Reviewer in that these values are too close to each other to draw clear conclusions. Given these considerations, as referred in the previous answer the text of the manuscript was changed in this section to reduce the possibility of incorrect interpretations.

-P14, L602. Given the discussion on alkenes vs epoxides, it would be interesting to see the activity of for instance intermediate 6.

Authors’ response: Thank you very much for this relevant commentary and suggestion. Indeed, presenting the activity of some intermediates would be an interesting approach. However, arylidene derivatives have been studied for many years and are well-known and described as antiproliferative agents in the literature. Thus, the focus of the present work was the synthesis and evaluation of steroidal epoxides considering its novelty and interest of the epoxide function, as described in the manuscript introduction.

-Figure 4: Caption should indicate what the arrows mean.

Authors’ response: The meaning of the arrows presented in Figure 4 was added to the respective legend.

-In references, please review the format of the doi for some references.

Authors’ response: Thank you. The DOI of some references were reviewed and corrected in the manuscript.

Reviewer 2 Report

In this manuscript, Samuel Silvestre et al “Synthesis, In Vitro Biological Evaluation of Antiproliferative and Neuroprotective Effects and In Silico Studies of Novel 16E-Arylidene-5a,6a-epoxyepiandrosterone Derivatives” authors reported as potent anti-cancer agents for the treatment of leukemia, breast and prostate cancers, and brain tumors with a sufficient data of their derivatives.   

I recognize the originality of the manuscript and its importance to the field, I recommended publishing this work after considering the below comments.

Comments to Authors:

1.      Compound-17 1H-NMR proton integration needs to be added.

2. Why authors did not mention the proton integration in between approximately 1.2 to 2.2 δ in ppm, for all derivatives?

3.      Did the authors study the toxic study of normal clines with these derivatives?

4.      Even though it is a well-established reaction from 2-14 to 15-27, did the authors try chiral HPLC supporting data for the asymmetric epoxy product derivatives?

5.      Why the compound 16 yield is very low (30%) for its conversion (3 to 16) compare to the other derivatives?

Author Response

REVIEWER#2

In this manuscript, Samuel Silvestre et al “Synthesis, In Vitro Biological Evaluation of Antiproliferative and Neuroprotective Effects and In Silico Studies of Novel 16E-Arylidene-5a,6a-epoxyepiandrosterone Derivatives” authors reported as potent anti-cancer agents for the treatment of leukemia, breast and prostate cancers, and brain tumors with a sufficient data of their derivatives.  

I recognize the originality of the manuscript and its importance to the field, I recommended publishing this work after considering the below comments.

Authors’ response: Thank you very much for your positive appreciation.

Comments to Authors:

  1. Compound-17 1H-NMR proton integration needs to be added.

Authors’ response: Thank you very much for this observation. This information was added in proton NMR spectrum for this compound presented in Supplementary material.

  1. Why authors did not mention the proton integration in between approximately 1.2 to 2.2 δ in ppm, for all derivatives?

Authors’ response: Thank you very much for your question. We understand your concern; however the signals of aliphatic protons without integration correspond to the so called “methylenic bloc”, which usually ranges from 0.5 to 2.5 ppm. These signals are originated from aliphatic protons without marked steric and/or electronic influence and thus have high overlapping. Considering this fact, it is only possible to perform the signals attribution to each proton with confidence using advanced NMR techniques and, in some cases, doubts will still persist. Therefore, it is frequent that in 1H-NMR steroids characterization only the signals outside the “methylenic bloc” as well as the signals with higher integration are presented. An example is the following article:

https://www.sciencedirect.com/science/article/pii/S0960076021001904?via%3Dihub#sec0010

Moreover, some authors also present these signals as “overlapping multiplets” integrating diverse numbers of hydrogens. As example, please see:

https://www.sciencedirect.com/science/article/pii/S0960076022000152?via%3Dihub

As we are presenting 1H-NMR spectra of these compounds in “Supplementary Material”, we believe that it is no necessary to include more information concerning this point in the manuscript.

  1. Did the authors study the toxic study of normal clines with these derivatives?

Authors’ response: Thank you for your question. Indeed, we studied the toxicity of these derivatives in non-tumoral cell lines, including NHDF (normal human dermal fibroblasts), PNT1A (human normal prostate epithelium cells) and N27 (rat dopaminergic neural cell line) cells.

  1. Even though it is a well-established reaction from 2-14 to 15-27, did the authors try chiral HPLC supporting data for the asymmetric epoxy product derivatives?

Authors’ response: Thank you very much for this pertinent question. In fact, we understand the potential limitation of using conventional NMR in determining the (enantiomeric) purity of compounds. However, at this level of drug discovery studies we believe that NMR data is enough to support the compounds purity and therefore we didn’t perform any study using HPLC to acquire data for the asymmetric epoxy product derivatives. Nevertheless, to proceed to more advanced preclinical studies with some of these molecules, clearly it will be necessary further purity analysis, including through (chiral) HPLC.

  1. Why the compound 16 yield is very low (30%) for its conversion (3 to 16) compared to the other derivatives?

Authors’ response: Thank you for this pertinent observation. A low yield was, in fact, observed in the epoxidation of steroid 3 (33 %); however, we believe that this yield is not related to reactivity questions since the substrate seemed completely consumed at the end of reaction (TLC observation). According to our observations, this discrepancy is probably associated with significant product losses during the workup procedure due to the fact that compound 16 was obtained as a very fine powder, allowing significant loss in filtration, even with refiltration. The product loss is also significant because reactions were performed with a low quantity of substrates. Despite this outcome, we didn’t invest more time and reagents in the improvement of this yield since the quantity obtained was enough to perform chemical characterization and biological assays.